# Crescents formations are independently associated with higher mortality in biopsy-confirmed immunoglobulin A nephropathy

**Cheng-Hsu Chen[1,2], Ming-Ju Wu[1], Mei-Chin Wen[3], Shang-Feng Tsai**📧[1,2,4]*

**1** Division of Nephrology, Department of Internal Medicine, Taichung Veterans General Hospital, Taichung, Taiwan, **2** Department of Life Science, Tunghai University, Taichung, Taiwan, **3** Department of Pathology, Taichung Veterans General Hospital, Taichung, Taiwan, **4** School of Medicine, National Yang-Ming University, Taipei, Taiwan

* s881056@gmail.com

**Data Availability Statement:** All relevant data are within the paper and its Supporting Information files.

## Abstract

### Background and objectives

Immunoglobulin A Nephropathy (IgAN) is the most common type of glomerulonephritis with variable renal outcome. The association between IgAN and patient survival is limited. The effect of crescents on patient survival was never studied.

### Materials

We conducted a retrospective cohort study between January 2003 and December 2013. All patients with the biopsy-proved IgAN was enrolled for the analysis of patient survival and renal survival. Cox regression model was used analyze the associated factors for patient survival.

### Results

All 388 participants with IgAN were enrolled, in which 45 patients with crescents. The mean percentage of glomeruli involvement was 23±18.9%. After long-term follow-up, crescents group had both worse renal (p = 0.034) and patient survivals (p = 0.016). In univariate Cox regression model, the age (HR = 1.08, 95% CI = 1.05–1.12, p<0.001), crescents (HR = 3.93, 95% CI = 1.18–13.07, p = 0.025), serum albumin (HR = 0.023, 95%CI = 0.11–0.50, p<0.001), blood total protein (HR = 0.46, 95%CI = 0.28–0.75, p = 0.002), HDL (HR = 0.95, 95%CI = 0.91–0.99, p = 0.009), daily urine protein (HR = 1.14, 95%CI = 1.01–1.29, p = 0.038), urine PCR (HR = 1.07, 95%CI = 1.02–1.12, p = 0.003), serum IgM (HR = 0.98, 95%CI = 0.96–1.00, p = 0.036), BUN (HR = 1.02, 95%CI = 1.01–1.02, p = 0.005), and eGFR (HR = 0.097, 95%CI = 0.94–0.99, p = 0.0011) were associated with patient survival. After multivariate Cox regression analysis, age (HR = 1.08, 95%CI = 1.01–1.13, p = 0.013), crescents (HR = 5.57, 95%CI = 1.14–29.05, p = 0.034), and HDL (HR = 0.94, 95%CI = 0.90–0.99, p = 0.026) were associated with patient survival. Crescents IgAN is with the highest risk (up to 5.75 of HR) for patient mortality.

**Funding:** This study was supported by Grants from Taichung Veterans General Hospital: TCVGH-1093605D and TCVGH-1093602B.

**Competing interests:** The authors have declared that no competing interests exist.

## Conclusions

The major strengths of the present study is that crescents IgAN had worse patient survival compared to non-crescents IgAN. Clinicians should be more careful to care patients with crescents IgAN.

## Introduction

Immunoglobulin A Nephropathy (IgAN) is the most common type of glomerulonephritis throughout the world [1, 2] with a wide range of histologic patterns and complex clinical manifestations. Even with standard treatment, IgAN still remains an important burden of end-stage renal disease (ESRD) [3]. Currently, certain clinical situations and histopathological features, including the degrees of glomerular and interstitial fibrosis, tubular atrophy, and glomerular hypercellularity are considered independent risk factors for the progression of IgAN to ERSD. The poor histological conditions can predict renal outcome. The Oxford Classification of IgAN does not include glomerular crescents before. Recently, there was a major update in the Oxford classification system [4]: crescents to the graded parameters, consisting of the MEST-C scores. This update was based on a large scale (3096 patients) performed by Haas et al in 2017 [4]. In that study [4], crescents in at least one sixth or one fourth of glomeruli associated with a significantly higher hazard ratio (HR) for a combined event of 1.63 or 2.29, respectively. Even the patients with crescents in less than 1/10 of biopsied glomeruli still showed worse composite renal outcome [adjusted HR 1.38 (95% CI 1.13–1.69), p = 0.002] than those without crescents [5]. The abovementioned findings all support that crescents is with clear association with renal outcome in patients with IgAN no matter the treatment. However, the association of crescents formation in IgAN and patients outcome is still unknown.

The association between IgAN and ESRD was much stronger than between IgAN and patient death [6, 7]. The patient survival in patients with IgAN had been studied more frequently recently. In a study in Norwegian in 2013, an age- and sex-adjusted mortality rate in Norwegian patients with IgAN is approximately twice that of the general Norwegian population [6]. However, previous studies were with limitations, including single center only [8], some IgAN subgroups [9, 10], and limited to IgAN already with ESRD [11]. Recently, a nationwide population-based cohort study was conducted in Sweden and the authors compared 3622 patients with IgAN with 18041 matched general population controls [12].They found a 53% relative increase in mortality and a modest increase in absolute death rate. On average, patients with IgAN died 6 years earlier than people without the disease. However, the association between detailed pathological findings of IgAN and patients mortality is still limited. Furthermore, the predictive value for patient mortality of crescents formation is also rare. Until now, there was only one study revealing that increasing crescents proportion was identified as an independent predictor for unfavorable clinical outcomes in IgAN [13]. However, the increased mortality in crescents IgAN was only based on indirect evidence and that patient outcome is only secondary outcome [4]. In IgAN with crescents, aggressive immunosuppressant may rescue renal function but the effect on patient outcome is still unknown. On the contrary, over-suppression of immune system could be harmful for patient survival. The outcome of crescents IgAN should be clarified first before our aggressive treatment. In our previous study, we found that a lower serum IgG ($\leq$907 mg/dL) and serum C3 ($\leq$79.7 mg/dL) were both risk factors for poor renal outcome in our database of IgAN [14]. Here, in this study, we

aimed to study the association between crescents formation and the patient mortality in IgAN in the same patients group [14].

## Materials and methods

### Study population

This retrospective cohort study was performed between January 2003 and December 2013. Participants of the age should be more than 20 years old and the diagnosis of IgAN was based on their first renal (native kidney only) biopsy in our medical center (Taichung Veterans General Hospital, TCVGH) in Taiwan. Our institute has possessed the largest patient population undergoing renal biopsies. Until now, the accumulative case numbers of renal biopsy was more than 8000. This study was approved by Ethics Committee of TCVGH, IRB number: CE15125B. All methods were performed in accordance with relevant guidelines and regulations and informed consent was obtained from all subjects.

### Data collection and outcome assessment

All data was obtained from this cohort via the reviewing of medical records. Baseline data while diagnosis was collected at the time of each patient's renal biopsy, including gender, age, body height (cm), body weight (kg), and systolic or diastolic blood pressure (mmHg) (SBP and DBP). Data from blood sample was also collected for blood urea nitrogen (BUN) (mg/dl), serum creatinine (mg/dl), estimated glomerular filtration rate (eGFR from Modification of Diet in Renal Disease equation) (ml/min/1.732m$^2$) [15], white blood cell (WBC) (/cumm), red blood cell (RBC) (/cumm), hemoglobin (g/dl), neutrophil percentage (%), platelet count (/cumm), uric acid (mg/dl), sodium (meq/L), potassium (meq/L), calcium (mg/dl), phosphate (mg/dl), magnesium (mg/dl), albumin (g/dl), total protein (g/dl), glutamate oxaloacetate transaminase (GOT) (U/L), glutamate-pyruvate transaminase (GPT) (U/L), total cholesterol (mg/dl), triglyceride (mg/dl), low-density lipoprotein (LDL) (mg/dl), high-density lipoprotein (HDL) (mg/dl), fasting and postprandial blood sugar (mg/dl), glycated hemoglobin (%). Chronic infection or inflammatory markers were included as follows; hepatitis B status, hepatitis C status, Anti-Nuclear Ab (ANA), anti-double stranded DNA (anti-dsDNA), Anti-neutrophil Cytoplasmic Antibodies (ANCAs), Proteinase 3 (PR3) and Myeloperoxidase (MPO). Urine samples were tested for daily urine proteinuria (g/day), urinary protein/creatinine ratio (PCR) (g/g) and urinary albumin/creatinine ratio (ACR) (mg/g).

All pathological samples were analyzed by an experienced pathologist while all enrolled participants had their diagnosis of IgAN based upon the criteria in the World Health Organization monograph of kidney disease [16]. The diagnosis of crescents IgAN was confirmed once with the report of crescents whatever the percentage of crescents formation. The proportion of crescents in biopsied tissue was also recorded as percentage of all glomeruli. The detailed pathological findings of IgAN were based on Oxford classification [17]. The study major outcome is patient death and secondary outcome is ESRD, whose who needed the initiation of dialysis, or those receiving transplantation according to local guidelines.

### Statistical methods

Data was expressed as the mean ± SD in continuous variables and as numbers (percentages) in categorical data. A Mann–Whitney U test was used for continuous variables and the *Chi*-square test was used for categorical variables. A Kaplan-Meier curve was implemented for measuring both patient survival and renal survival. A Cox proportional hazard regression (shown as HR, 95% confidence interval (CI)) was used to analyze the possible factors for

patient survival (both the univariate and multivariate Cox models). Initially in the univariate analysis, all possible factors were analyzed. Then we chose each possible associated factors from each associated categories (such as baseline physical condition (age, and gender), pathological data for IgAN (crescents formation), blood laboratory data for IgAN (secondary hyperlipidemia, low albumin, low total protein, low LDL and low HDL), urinary laboratory data for IgAN (daily urine protein, and urinary PCR) and data for renal function evaluation (serum creatinine, BUN, and eGFR)), which were significantly associated factors in univariate analysis, to be analyzed in multivariate Cox regression model.

A value of $p < 0.05$ was considered statistically significant. All statistical procedures were performed using the SPSS statistical software package, version 17.0 (Chicago, IL).

## Results

All 388 participants (age > 20 years old) were enrolled in this study because of the diagnosis of IgAN between January 2003 and December 2013. Forty-five IgAN-patients were with crescents whatever the percentage of glomeruli involvement. The whole duration of follow-up was 11 years (January 2003 to December 2013). The mean duration of follow-up was 7.2 ± 3.1 years. In Table 1, patients with crescents IgAN were more stages of E, S, and T scores of Oxford classification, hypocalcemia (p = 0.001), hypoalbuminemia (p<0.001), higher LDL (p = 0.002), lower HDL (p = 0.047), more daily urine protein (p = 0.005), higher urinary PCR (p<0.001), higher blood IgG (p = 0.034), lower IgM (p = 0.035), and more positive ANCA (p<0.001). As for the crescents IgAN (S1 Table), the mean percentage of glomeruli involvement was 23 ±18.9% (medium percentage of crescents IgAN is 18%). Only 4 (8.9%) patient had more than 50% crescents formation of IgAN (S2 Table and S1 Fig).

The 1-year, 3-year and 5-year renal survival were 98.4% vs. 92.8%, 93.4% vs. 89.8%, and 86.7% vs. 77.3% in non-crescents IgAN and crescents IgAN (Fig 1). After long-term follow-up, the renal survival was better in patients with IgAN without crescents formation (p = 0.034). As for patient survival (Fig 2), the 1-year, 3-year and 5-year renal survival were 98.8 vs. 95.2%, 97.8 vs. 95.2%, and 97.1 vs. 86.5% in non-crescents IgAN and crescents IgAN. After long-term follow-up, the patient survival was better in patients with IgAN without crescents formation (p = 0.016). The infection related cause of death was 62.5% and 75% in non-cresent IgAN and cresent IgAN (S3 Table).

The possible variables associated with patient survival were list in Table 2. In univariate Cox regression model, the age (HR = 1.08 (95% CI = 1.05–1.12), p<0.001), crescents formation (HR = 3.93 (95% CI = 1.18–13.07), p = 0.025), blood albumin (HR = 0.023 (95% CI = 0.11–0.50), p<0.001), blood total protein (HR = 0.46 (95% CI = 0.28–0.75), p = 0.002), HDL (HR = 0.95 (95% CI = 0.91–0.99), p = 0.009), daily urine protein (HR = 1.14 (95% CI = 1.01–1.29), p = 0.038), urine PCR (HR = 1.07 (95% CI = 1.02–1.12), p = 0.003), serum IgM (HR = 0.98 (95% CI = 0.96–1.00), p = 0.036), BUN (HR = 1.02 (95% CI = 1.01–1.03), p = 0.005), and eGFR (HR = 0.097 (95% CI = 0.94–0.99), p = 0.0011) were associated with patient survival. After multivariate Cox regression analysis, older age (HR = 1.08 (95% CI = 1.01–1.13), p = 0.013), crescents (HR = 5.57 (95% CI = 1.14–29.05), p = 0.034), and HDL (HR = 0.094 (95% CI = 0.90–0.99), p = 0.026) were associated with patient survival. Crescents IgAN is with the highest risk (up to 5.75 of HR) for patient mortality.

## Discussion

In this study, focusing on the effect of crescents of IgAN on patient survival revealed that mortality was increased independently from other traditional risk factors for patient mortality. According to previous studies [11, 12], IgAN is associated with more hypertension and

**Table 1. Basic characteristics of IgA nephropathy based on crescents or non-crescents.**

| | Total (n = 388) | Non Crescents(n = 343) | Crescents (n = 45) | *p* value |
|---|---|---|---|---|
| Sex: female | 175 (45.1%) | 157 (45.8%) | 18 (40.0%) | 0.567 |
| Age (years old) | 40.91±15.39 | 40.85±15.14 | 41.40±17.42 | 0.970 |
| Oxford classification (n, %) | | | | |
| M0 | 247 | 223 (90.3%) | 24 (53.3%) | 0.078 |
| M1 | 138 | 118 (34.4%) | 20 (44.4%) | 0.087 |
| M2 | 3 | 2 (0.6%) | 1 (2.2%) | 0.227 |
| E0 | 227 | 296 (86.3%) | 31 (68.9%) | 0.04 |
| E1 | 62 | 48 (14.0%) | 14 (31.1%) | 0.002 |
| S0 | 341 | 322 (93.9%) | 19 (45.2%) | <0.001 |
| S1 | 47 | 21 (6.1%) | 26 (57.8%) | <0.001 |
| T0 | 332 | 298 (86.9%) | 34 (75.6%) | 0.073 |
| T1 | 38 | 29 (8.5%) | 9 (20%) | 0.028 |
| T2 | 18 | 16 (4.7%) | 2 (4.4%) | 0.788 |
| C0 | 388 | 343 (100%) | 0 (0%) | <0.001 |
| C1 | 35 | 0 | 35 (77.8%) | <0.001 |
| C2 | 10 | 0 | 10 (22.2%) | <0.001 |
| SBP (mmHg) | 133.43±19.11 | 133.88±18.81 | 130.00±21.17 | 0.250 |
| DBP (mmHg) | 84.20±14.03 | 84.45±13.61 | 82.33±16.90 | 0.616 |
| Body height (cm) | 102.61±48.92 | 102.36±49.25 | 104.51±46.78 | 0.330 |
| Body weight (kg) | 101.09±49.19 | 101.71±49.57 | 96.29±46.39 | 0.944 |
| Blood data | | | | |
| Blood urea nitrogen (mg/dl) | 31.14±26.61 | 30.43±26.43 | 36.51±27.61 | 0.150 |
| Creatinine (mg/dl) | 2.43±3.15 | 2.37±3.18 | 2.94±2.93 | 0.112 |
| eGFR (ml/min/1.732m$^2$) | 57.15±40.00 | 58.05±40.20 | 50.10±38.15 | 0.190 |
| Blood WBC (/μL) | 8016.25±2898.53 | 7928.69±2911.87 | 8681.78±2734.85 | 0.076 |
| Hemoglobin (g/dl) | 12.16±2.23 | 12.20±2.20 | 11.90±2.50 | 0.715 |
| Neutrophil (%) | 65.34±10.78 | 65.09±10.56 | 67.28±12.32 | 0.247 |
| Platelet (x10$^3$/μL) | 248.05±90.16 | 244.01±73.29 | 278.78±169.07 | 0.336 |
| Uric acid (mg/dl) | 7.45±2.19 | 7.39±2.14 | 7.86±2.48 | 0.285 |
| Sodium (meq/L) | 139.76±3.49 | 139.81±3.54 | 139.38±3.12 | 0.195 |
| Potassium (meq/L) | 4.28±0.55 | 4.27±0.56 | 4.37±0.51 | 0.249 |
| Calcium (mg/dl) | 8.51±1.07 | 8.57±1.02 | 8.08±1.33 | 0.001** |
| Phosphate (mg/dl) | 4.00±1.29 | 3.97±1.25 | 4.26±1.54 | 0.223 |
| Magnesium (mg/dl) | 2.28±0.46 | 2.26±0.46 | 2.37±0.44 | 0.490 |
| Albumin (g/dl) | 3.69±0.65 | 3.73±0.64 | 3.37±0.62 | <0.001** |
| Total protein (g/dl) | 6.54±0.94 | 6.59±0.92 | 6.19±0.97 | 0.010* |
| GOT (U/L) | 23.62±19.46 | 23.84±20.25 | 22.00±12.25 | 0.783 |
| GPT (U/L) | 23.94±22.17 | 23.87±22.47 | 24.49±20.05 | 0.693 |
| Total cholesterol (mg/dl) | 193.50±54.59 | 192.17±54.25 | 204.26±56.87 | 0.062 |
| Triglyceride (mg/dl) | 153.31±136.16 | 149.52±132.02 | 184.24±165.00 | 0.140 |
| Low-density lipoprotein(mg/dl) | 117.21±43.30 | 114.21±39.49 | 141.54±62.12 | 0.002** |
| High-density lipoprotein | 55.02±19.52 | 55.60±19.31 | 49.64±21.03 | 0.047* |
| Fasting glucose (mg/dl) | 95.70±21.20 | 96.16±22.08 | 92.19±12.37 | 0.552 |
| Postprandial glucose (mg/dl) | 143.40±66.99 | 143.68±67.36 | 140.00±76.22 | 1.000 |
| Glycated hemoglobin (%) | 5.81±1.15 | 5.83±1.20 | 5.66±0.63 | 0.567 |
| IgG (mg/dl) | 1093.09±342.02 | 1103.93±326.09 | 1011.76±440.59 | 0.034* |
| IgA (mg/dl) | 353.51±151.70 | 352.20±151.35 | 363.34±155.82 | 0.588 |

(*Continued*)

**Table 1.** (Continued)

| | Total (n = 388) | Non Crescents(n = 343) | Crescents (n = 45) | *p* value |
|---|---|---|---|---|
| IgM (mg/dl) | 111.20±53.12 | 113.09±53.16 | 97.08±51.30 | 0.035* |
| IgE (mg/dl) | 256.94±563.22 | 252.09±576.56 | 290.90±476.11 | 0.765 |
| C3 (mg/dl) | 109.40±24.66 | 108.93±24.29 | 112.61±27.12 | 0.368 |
| C4 (mg/dl) | 29.56±10.14 | 29.52±10.10 | 29.83±10.53 | 0.867 |
| HBsAg positive | 10 (18.5%) | 8 (20.5%) | 2 (13.3%) | 0.398 |
| AntiHBs positive | 17 (40.5%) | 11 (36.7%) | 6 (50.0%) | 0.735 |
| AntiHCV positive | 11 (3.2%) | 11 (3.6%) | 0 (0.0%) | 0.384 |
| ANA positive | 31 (8.7%) | 25 (8.0%) | 6 (13.3%) | 0.500 |
| dsDNA positive | 19.98±17.88 | 19.76±17.09 | 21.15±22.02 | 0.732 |
| ANCA positive | 6 (2.8%) | 1 (0.5%) | 5 (15.6%) | <0.001** |
| Myeloperoxidase | 10.16±27.08 | 2.30±1.96 | 28.21±45.58 | 0.070 |
| Proteinase-3 | 3.47±3.14 | 2.88±2.54 | 5.11±4.14 | 0.150 |
| Rapid plasma reagin | 1 (0.8%) | 1 (0.9%) | 0 (0.0%) | 1.000 |
| Daily urine protein (g/day) | 2.40±2.78 | 2.28±2.74 | 3.36±2.89 | 0.005** |
| Urinary protein/creatinine ratio (g/g) | 2.45±4.26 | 2.28±4.24 | 3.74±4.18 | <0.001** |
| Urinary albumin creatinine ration (mg/g) | 778.37±1159.31 | 728.21±1146.22 | 1254.96±1253.97 | 0.101 |

Chi-square test. Mann-Whitney U test.

*p<0.05,

**p<0.01.

vascular disease after long-term follow-up. Investigators also found a 59% increased risk for death from cardiovascular disease (CVD) [12]. Another study also showed that 45% of all death in IgAN was due to CVD [6]. Both decreased eGFR and increased proteinuria increase the risk of CVD from many studies [18–21]. Once the IgAN cannot be cured, all patients will go into chronic kidney disease (CKD). CKD was also considered as a coronary heart disease risk equivalent based on several studies [22–24]. Besides, as CKD progression, the CV outcome also got worse, including death from CV causes, re-infarction, congestive heart failure, stroke, resuscitation and composite end points [25]. In our study, the baseline eGFRs between non-crescents and crescents group were similar (58.05 vs. 50.10 ml/min/1.732m$^2$, p = 0.190). However, after long-term follow-up of renal function, renal function was significantly worse in crescents group than non-crescents group (p = 0.034). Therefore, crescents group went into more advanced stage of CKD and would experience more CVD. Moreover, worse controlled IgAN will have more severe secondary hyperlipidemia, which further predisposed more CVD after long-term follow-up.

In addition to worse renal function related CV complication in crescents IgAN, increased mortality in patients treated with steroids or immunosuppressive agents was also reported in IgAN from other study [12]. In a prematurely terminated trial to evaluate the efficacy and safety of steroids in patients with IgAN with more than 1g/day proteinuria, oral methylprednisolone was associated with an increased risk of serious infections [26]. Rauen et al conducted a multicenter, open-label, randomized, controlled study that more adverse effects were observed among the patients who received immunosuppressive therapy, with no change in the rate of decrease in the eGFR [27]. Another randomized controlled trial showed that after a median of 2.1 years, serious infection occurred in 14.7% percent of patients who received methylprednisolone compared with 3.2 percent of those who received placebo [26]. Crescents IgAN tend to receive more immunosuppressant then non-crescents group. Reasonably,

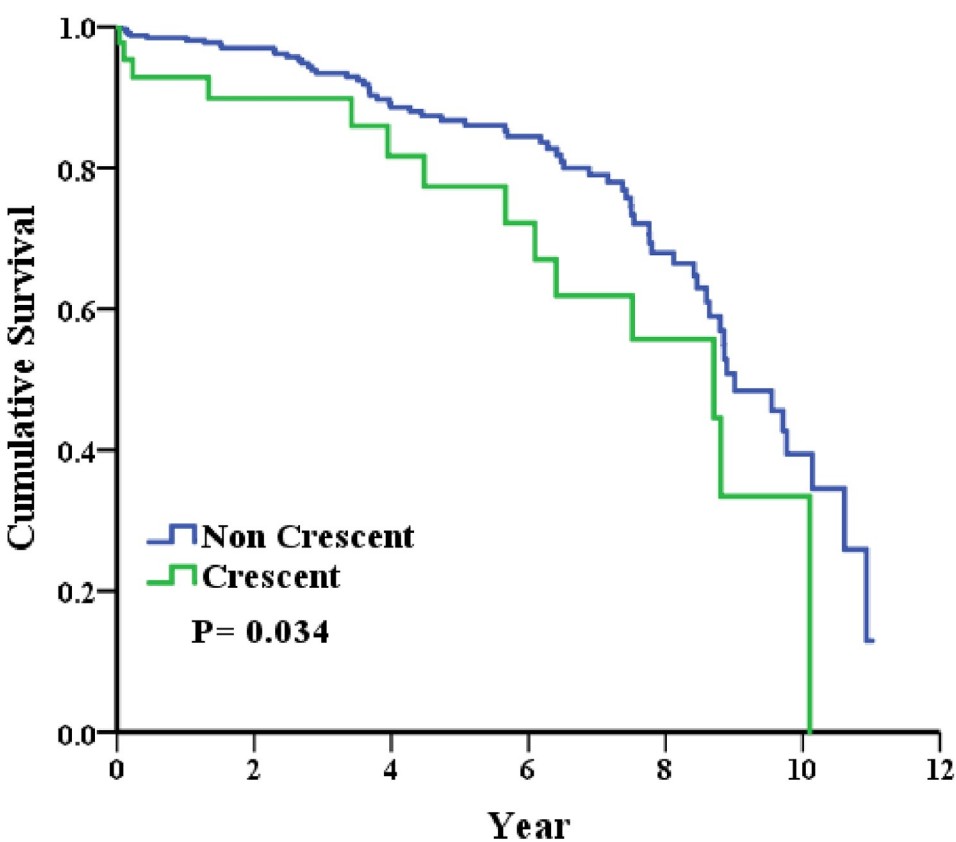

Overall survival

| | Total | HD | Non-HD | Non-HD Percent | Number at risk (n) and survival rate (%) | | | p for log rank |
|---|---|---|---|---|---|---|---|---|
| | | | | | 1y | 3y | 5y | |
| Non Crescent | 343 | 63 | 280 | 81.6% | 337, 98.4% | 320, 93.4% | 297, 86.7% | 0.034 |
| Crescent | 45 | 14 | 31 | 68.9% | 42, 92.8% | 40, 89.8% | 35, 77.3% | |
| Overall | 388 | 77 | 311 | 80.2% | | | | |

**Fig 1. Real survivals based on crescent or non-crescent.**

patients with crescents IgAN would experience more infection and infection related death. That was consistent with our data (75% vs. 62.5%).

Apart from the above reasons, more crescents IgAN went into ESRD and received hemodialysis or peritoneal dialysis. They were prone to experience dialysis related complications, including CVD and infection. From a nation-wide stud, a strongly increased mortality (HR = 4.9) is observed in IgAN undergoing dialysis [6]. Our data also showed similar result that the patient survival between no crescents and crescents differ more significantly after longer follow-up (3.6%, 4.6%, and 10.6% for 1-year, 3-year and 5-year, respectively).

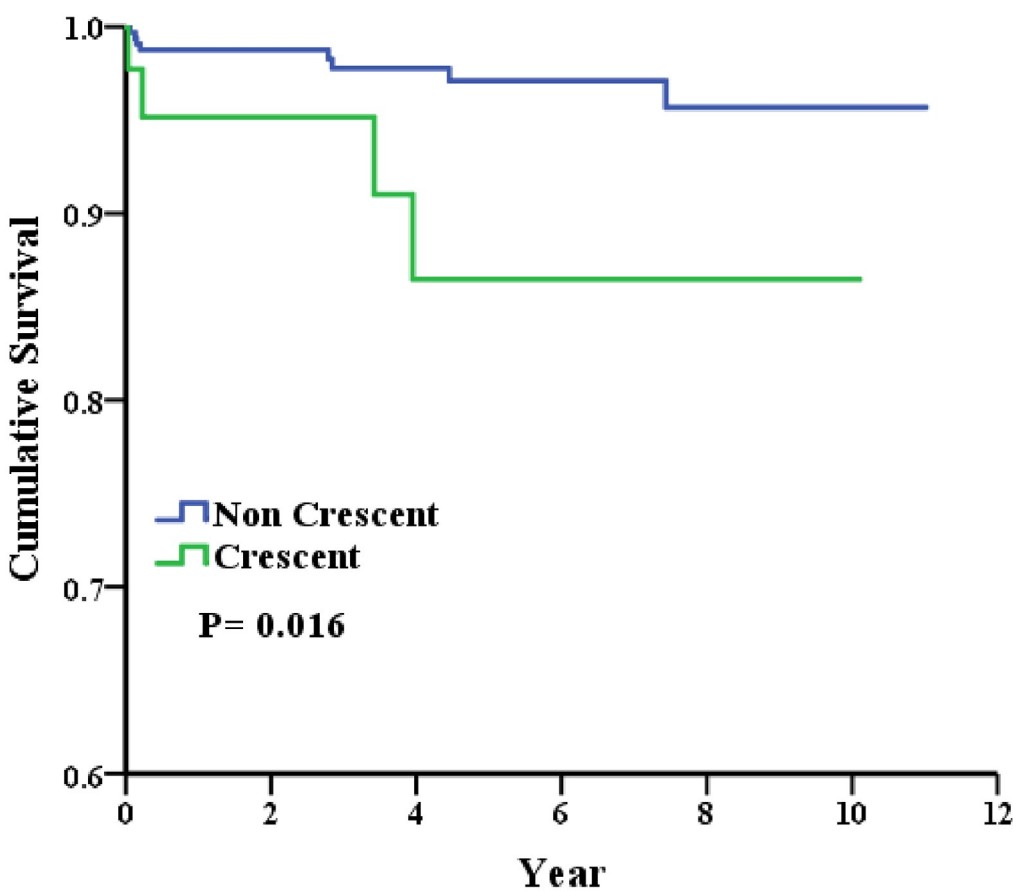

Overall survival

| | Total | Dead | Alive | Alive Percent | Number at risk (n) and survival rate (%) | | | p for log rank |
|---|---|---|---|---|---|---|---|---|
| | | | | | 1y | 3y | 5y | |
| Non Crescent | 343 | 8 | 335 | 97.7% | 339, 98.8% | 335, 97.8% | 333, 97.1% | 0.016 |
| Crescent | 45 | 4 | 41 | 91.1% | 43, 95.2% | 43, 95.2% | 39, 86.5% | |
| Overall | 388 | 12 | 376 | 96.9% | | | | |

**Fig 2. Patient survival based on crescent or non-crescent.**

In 2012, a Japanese prognostic model was validated in a Norwegian cohort [28]. In that scoring system, only one variable was histological score (5 scores if grade 3 or 4). However, no detailed analysis of crescents was performed. A single-center study following 30 years for the mortality of IgAN showed that more severe in both glomerular and tubulointerstitial areas in ESRD patients and in those who died [8]. However, the effect of crescents on mortality was not studied. Another study conducted by Knoop et al in 2013 showed increased mortality if baseline eGFR< 60 ml/min/1.732m$^2$ or DUP≧3g [6]. Our study also supports this risk

**Table 2. Univariate and multivariate Cox regression analyses on patient survivals.**

|  | Univariate | | | Multivariate | | |
|---|---|---|---|---|---|---|
|  | Hazard ratio | 95%CI | *p* value | Hazard ratio | 95%CI | *p* value |
| Sex |  |  |  |  |  |  |
| Female | Reference | Reference |  |  |  |  |
| Male | 4.44 | (0.97–20.25) | 0.055 |  |  |  |
| Age | 1.08 | (1.05–1.12) | <0.001** | 1.07 | (1.01–1.13) | 0.013* |
| Crescents |  |  |  |  |  |  |
| 0 | Reference | Reference |  | Reference | Reference |  |
| 1 | 3.93 | (1.18–13.07) | 0.025* | 5.75 | (1.14–29.05) | 0.034* |
| Albumin | 0.23 | (0.11–0.50) | <0.001** | 0.38 | (0.05–3.09) | 0.367 |
| Total protein | 0.46 | (0.28–0.75) | 0.002** | 0.85 | (0.22–3.22) | 0.809 |
| Low-density lipoprotein | 0.99 | (0.97–1.00) | 0.154 |  |  |  |
| High-density lipoprotein | 0.95 | (0.91–0.99) | 0.009** | 0.94 | (0.90–0.99) | 0.026* |
| Daily urine protein | 1.14 | (1.01–1.29) | 0.038* | 0.97 | (0.78–1.20) | 0.771 |
| Urinary protein/creatinine ratio | 1.07 | (1.02–1.12) | 0.003** |  |  |  |
| IgG | 1.00 | (1.00–1.00) | 0.398 |  |  |  |
| IgM | 0.98 | (0.96–1.00) | 0.036* | 0.99 | (0.97–1.02) | 0.453 |
| IgA | 1.00 | (1.00–1.00) | 0.075 |  |  |  |
| C3 | 0.98 | (0.95–1.00) | 0.094 |  |  |  |
| C4 | 1.02 | (0.97–1.07) | 0.447 |  |  |  |
| Blood urea nitrogen | 1.02 | (1.01–1.03) | 0.005** |  |  |  |
| Creatinine | 1.06 | (0.96–1.17) | 0.277 |  |  |  |
| eGFR | 0.97 | (0.94–0.99) | 0.011* | 1.00 | (0.96–1.03) | 0.915 |
| ANA |  |  |  |  |  |  |
| Negative | Reference | Reference |  |  |  |  |
| Positive | 1.56 | (0.19–12.84) | 0.678 |  |  |  |
| ANCA |  |  |  |  |  |  |
| Negative | Reference | Reference |  |  |  |  |
| Positive | 7.59 | (0.88–65.49) | 0.065 |  |  |  |

Cox proportional hazard regression.

*$p < 0.05$,

**$p < 0.01$.

prediction to mortality. As for eGFR, both crescents (50.10 ml/min/1.732m$^2$) and non-crescents (58.05 ml/min/1.732m$^2$) group are moderate risk group according to that study [8]. As for proteinuria, crescents group is high risk (3.36 g/day) group but non-crescents group is moderate risk group (2.28 g/day). Our data further support their results [8] based on new pathological evidence.

Our study is similar to a previous study [13], conducted by Zhang et al in 2017. In that study, authors found that less than 50% crescents involvement was associated with worse renal outcome without statistical significance (p = 0.077). The composite outcome (renal and patient survival) showed significant association (p = 0.003). Multivariate Cox regression analyses adjusting for eGFR, hypertension, proteinuria, and the Oxford-MEST classification demonstrated the predictive significance of an increasing crescents proportion with composite outcome (HR = 1.51). However, there was no direct analysis (Kaplan–Meier curves and Multivariate Cox regression analysis) for patient survival in that study [13]. They just showed renal outcome and incorporated (renal and patient survival) outcome. It is worth mentioning

that our study is the first one focusing on patient survival of IgAN with crescents. In summary, crescents formation in IgAN is associated with higher mortality in our study with direct evidence and in previous study with indirect evidence [13]. As for renal survival, the 5-your renal outcome was worse in our cohort compared to that study (77.3 vs. 86.9%) [13]. That reason was as follows. Comparing to that cohort, our patients with crescents were with older age (41.4 vs. 32 years old), more severe baseline renal function (50.1 vs. 84 ml/min/1.732m2 of eGFR, and 3.36 vs. 0.8 g/day of DUP) and more severe crescents involvement (6.7, 17.8, 37.8 and 37.8% vs. 28.0, 31.2, 32.7, and 8.0%, for <5%, 5–9%, 10–24%, and ≥ 25% crescents, respectively).

There are some limitations for this study. First, the case number of this study is relatively low. Second, we did not include the treatment for IgAN (especially the dose of corticosteroid). However, because this study was conducted in single institute, the treatment protocol is almost the same. For IgAN without crescents, steroid based therapy was applied. As for IgAN with crescents, we always applied induction therapy with therapeutic plasmapheresis for five times and methylprednisolone 500mg for three days. All patients with IgAN should receive renin–angiotensin–aldosterone system blockader as possible. Third, we did not evaluate the association between detailed MEST-C scores and patient outcome. However, the main object of this study focused on crescents and all other histological parameters for patient outcome had been already evaluated in previous studies. Fourth, we did not exclude 6 patients with ANCA+ and IgAN. That was because they did have systemic manifestation of vasculitis and they were not diagnosed as pauci-immune glomerulonephritis according to immunofluorescence microscopy. Finally, we did not have information of blood pressure and other disease after long-term follow-up.

## Conclusion

The major strengths of the present study is that crescents IgAN had worse patient survival compared to non-crescents IgAN. Clinicians should be more careful to care patients with crescents IgAN.

## Supporting information

**S1 Table. All patients with crescents IgAN.**
(DOCX)

**S2 Table. Distribution of the proportions of crescents.**
(DOCX)

**S3 Table. Cause of death.**
(DOCX)

**S1 Fig. Distribution of the proportions of glomeruli with crescents.**
(DOCX)

## Acknowledgments

The authors thank the Biostatistics Task Force of Taichung Veterans General Hospital and Mr. Chen, Jun-Peng for help in statistics.

## Author Contributions

**Conceptualization:** Cheng-Hsu Chen, Shang-Feng Tsai.

**Data curation:** Cheng-Hsu Chen, Ming-Ju Wu, Shang-Feng Tsai.

**Formal analysis:** Cheng-Hsu Chen, Mei-Chin Wen.

**Funding acquisition:** Ming-Ju Wu, Shang-Feng Tsai.

**Investigation:** Cheng-Hsu Chen.

**Methodology:** Cheng-Hsu Chen.

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
