## [Decision Letter · Decision Letter 0]

12 Jun 2020

PONE-D-20-15033

Crescent formations are independently associated with higher mortality in biopsy-confirmed immunoglobulin A nephropathy

PLOS ONE

Dear Dr. Tsai,

Thank you for submitting your manuscript to PLOS ONE. After careful consideration, we feel that it has merit but does not fully meet PLOS ONE’s publication criteria as it currently stands. Therefore, we invite you to submit a revised version of the manuscript that addresses the points raised during the review process.

Please ensure to address all the point raised by reviewer 1  and reviewer 2.

We look forward to receiving your revised manuscript.

Kind regards,

Fabio Sallustio

Academic Editor

PLOS ONE

Journal Requirements:

2. Please provide additional details regarding participant consent. In the ethics statement in the Methods and online submission information, please ensure that you have specified what type of consent you obtained (for instance, written or verbal, and if verbal, how it was documented and witnessed).

https://www.mdpi.com/2077-0383/8/6/848/html

The text that needs to be addressed involves the Introduction.

In your revision ensure you cite all your sources (including your own works), and quote or rephrase any duplicated text outside the methods section. Further consideration is dependent on these concerns being addressed.

"This study was supported by Grants from Taichung Veterans General Hospital:

TCVGH-1093605D and TCVGH-1093602B."

Reviewers' comments:

Reviewer's Responses to Questions

**Comments to the Author**

1. Is the manuscript technically sound, and do the data support the conclusions?

Reviewer #1: Partly

Reviewer #2: Yes

2. Has the statistical analysis been performed appropriately and rigorously? 

Reviewer #1: N/A

Reviewer #2: Yes

3. Have the authors made all data underlying the findings in their manuscript fully available?

Reviewer #1: No

Reviewer #2: Yes

4. Is the manuscript presented in an intelligible fashion and written in standard English?

Reviewer #1: No

Reviewer #2: Yes

5. Review Comments to the Author

Reviewer #1: Chen et al have studied the effect of glomerular crescents in 388 biopsy-proven IgA nephropathy (IgAN) patients included in a retrospective study with a mean follow-up of 7.3±3.1 years. Crescents were present in 12% of cases. Results demonstrated that age, crescents and HDL were associated with patient survival at multivariate Cox regression analysis. Moreover, crescents expressed the highest risk for patient mortality.

The paper has many scientific drawbacks:

1. Kidney biopsies were not analyzed according to the Oxford Classification (MEST-C).

The number of crescents and their type (proliferative, fibrous) are not reported. This referee suggests to follow the paper of M. Haas et al on the crescents in IgAN kidney biopsies.

The percentage of florid crescents is not reported.

The percentage of tubulo-interstitial lesions is not reported. These lesions are responsible for the prognosis and outcome of the disease.

2. The Haas classification is out of date because the international Oxford classification is used by all pathologists.

3. ANCA positivity was present in 15.6% of IgAN cases with crescents. This means that these cases are not idiopathic IgAN. Therefore, the authors should exclude these patients.

4. Fig 1 and 2 are not correct. The number of cases observed at each time for the two subgroups is not reported.

5. The therapy should be described in details in material and methods. How many patients received plasmapheresis?

6. Dosage of corticosteroids is not reported.

7. Conclusions on aggressive immunosuppressive therapy are not supported by clinical data in the study that shows many lacking points.

8. The authors should evaluate the number of crescents; therefore, the term crescent is not appropriate and should be replaced by the term crescents.

Minor points

1. Reference No.1 is out of date because there are recent published papers on the frequency of IgAN in the world

2. P9 line 13 controlled trial

3. P10 eGFR>60 ml/min/1.732m2; should be eGFR<60 ml/min/1.732m2

Reviewer #2: I have no competing interests.

Interesting study with some limitations.

Abstract: why authors in the conclusion spoke about "Aggressive immunosuppressants should not

be given for all crescent IgAN and should be individualized". This does not seem to have relation with reported results

Abstract. CI should be added for all dependent variables

Methods: normality should be checked for

Methods: levels of significance for values inserted in multvariate analysis should be added

Methods/results: ratio of n of events/n of variavbles seems to low (see PMID: 8970487)

6. PLOS authors have the option to publish the peer review history of their article (what does this mean?). If published, this will include your full peer review and any attached files.

Reviewer #1: No

Reviewer #2: Yes: Fabrizio D'Ascenzo

---

## [Author Response · Author response to Decision Letter 0]

13 Jun 2020

Editor comments

Ok. 

2. Please provide additional details regarding participant consent. In the ethics statement in the Methods and online submission information, please ensure that you have specified what type of consent you obtained (for instance, written or verbal, and if verbal, how it was documented and witnessed).

 All methods were carried out in accordance with relevant guidelines and regulations and informed consent was obtained from all subjects.

https://www.mdpi.com/2077-0383/8/6/848/html

The text that needs to be addressed involves the Introduction.

We addressed that in the part of introduction. 

“In our previous study, we found that a lower serum IgG (≤907 mg/dL) and serum C3 (≤79.7 mg/dL) were both risk factors for poor renal outcome in our database of IgAN[J Clin Med

. 2019 Jun 14;8(6):848] ”

In your revision ensure you cite all your sources (including your own works), and quote or rephrase any duplicated text outside the methods section. Further consideration is dependent on these concerns being addressed.

We rephrased that. Thanks for your comments.

“This We conducted a retrospective cohort study was performed between January 2003 and December 2013. Participants of the age should be more than > 20 years old and the diagnosis of IgAN was based on who had undergone their first renal (native kidney only) biopsy with the diagnosis of IgAN were enrolled in our medical center (Taichung Veterans General Hospital, TCVGH) in Taiwan. Graft renal biopsies were excluded. Our institute has This medical center possessesd the largest patient population undergoing renal biopsiesfor those who have undergone renal biopsies (m. Until now, the accumulative case numbers of renal biopsy was more than ore than 8000 within the last 30 years). This study was approved by Ethics Committee of TCVGHTaichung Veterans General Hospital, IRB number：CE15125B. All methods were carried outperformed in accordance with relevant guidelines and regulations and informed consent was obtained from all subjects. ”

Please remove any funding-related text from the manuscript and let us know how you would like to update your Funding Statement. 

We removed that. 

We cite all supplementary data in the text. Also, we added the information of supplementary data at the end of our manuscript (as follows).

Supplementary data

Table 1. All patients with crescent IgAN.

Table 2. Distribution of the proportions of crescents 

Table 3. Cause of death 

Figure 1 Distribution of the proportions of glomeruli with crescents

Reviewer #1

1. Kidney biopsies were not analyzed according to the Oxford Classification (MEST-C).

The number of crescents and their type (proliferative, fibrous) are not reported. This referee suggests to follow the paper of M. Haas et al on the crescents in IgAN kidney biopsies. The percentage of florid crescents is not reported. The percentage of tubulo-interstitial lesions is not reported. These lesions are responsible for the prognosis and outcome of the disease.

Thanks for this comment. The focus of this article is impact of the presence of crescent formation on patients’ survival. And our result showed that the significant association between crescent formation and mortality. We did not aim to elucidate the type of crescent formation (proliferative, fibrous), which maybe more associated with renal outcome, rather than patient outcome. Similarly, our goal is this study is not to research the tubular or interstitial involvement of IgAN. Besides, the typical finding of IgAN is on glomerular change, rather than tubular or interstitial change. Thus, we did not show the detailed pathological finding in our study. 

2. The Haas classification is out of date because the international Oxford classification is used by all pathologists.

Thanks for this comment. We also agreed that Oxford classification is used by most pathologists. There have been 22 validation studies of the Oxford classification [Curr Opin Nephrol Hypertens 2017, 26, 165-171]. Recently, Park el al suggested that the Haas and the Oxford classifications are comparable in predicting the progression of IgAN [Human pathology 2014, 45, 236-243]. In our previous study [J Clin Med . 2019 Jun 14;8(6):848.], we proved that Haas classification is also useful for establishing predictive values in Asian groups. Therefore, the validation of Haas classification in our IgAN cohort had been done in our previous study. In addition, our aim of this study is crescent formation or not in IgAN. Not only Oxford but also Haas classification can show crescent formation. So, types of classification of IgAN do not affect the study design or outcome.

3. ANCA positivity was present in 15.6% of IgAN cases with crescents. This means that these cases are not idiopathic IgAN. Therefore, the authors should exclude these patients. 

Totally, there are only 6 patients (2.8%) with ANCA+ in this IgAN cohort. It is debatable whether IgG-ANCA plays a pathogenic role in IgAN. We did not exclude this ANCA+ IgAN patents because of following reasons. First, in this study, we enrolled patients with IgAN (including idiopathic or not, primary or secondary type). Therefore, we did not mention in the text that our patients are with idiopathic IgAN. Second (more important), all 6 patients were not diagnosed as ANCA related crescentic GN because of not pauci-immune finding in immunofluorescence microscopy. Third, these 5 patients with ANCA+ IgAN did not have systemic manifestation of vasculitis. Based on the above reasons, we did not exclude these 6 patients in this study. We mentioned this in the part of limitation, page 11. 

4. Fig 1 and 2 are not correct. The number of cases observed at each time for the two subgroups is not reported.

Thanks for this comment. We only showed the rate (%). As your suggestion, we added the information of “Number at risk (n) and survival rate (%)”

5. The therapy should be described in details in material and methods. How many patients received plasmapheresis?

Thanks for this comment. That is because we did not have detailed information of therapy of crescent IgAN. We mentioned this limitation in the part of limitation, page 11. However, because this study was conducted in single institute, the treatment protocol is almost the same. For IgAN without crescent, steroid based therapy was applied. As for IgAN with crescent, we always applied induction therapy with therapeutic plasmapheresis for five times (all patients with crescent IgAN) and methylprednisolone 500mg for three days. All patients with IgAN should receive renin–angiotensin–aldosterone system blockader as possible. We acknowledged this limitation.

6. Dosage of corticosteroids is not reported.

Thanks for this comment. That is because we did not have detailed information of therapy of crescent IgAN. We mentioned this limitation in the part of limitation, page 11. However, because this study was conducted in single institute, the treatment protocol is almost the same. For IgAN without crescent, steroid based therapy was applied. As for IgAN with crescent, we always applied induction therapy with therapeutic plasmapheresis for five times (all patients with crescent IgAN) and methylprednisolone 500mg for three days. All patients with IgAN should receive renin–angiotensin–aldosterone system blockader as possible. We acknowledged this limitation. The information regarding doses of steroid is lacking. We emphasize this in the part of limitation, page 11. 

“Second, we did not include the treatment for IgAN (especially the dose of corticosteroid)”

7. Conclusions on aggressive immunosuppressive therapy are not supported by clinical data in the study that shows many lacking points.

Thanks for this comment. We revised the part of conclusion as follows. “The major strengths of the present study is that crescent IgAN had worse patient survival compared to non-crescent IgAN. Clinicians should be more careful to care patients with crescent IgAN.

”

8. The authors should evaluate the number of crescents; therefore, the term crescent is not appropriate and should be replaced by the term crescents.

We replaced crescent with crescents. Thanks for your suggestion. 

Minor points

1. Reference No.1 is out of date because there are recent published papers on the frequency of IgAN in the world

Thanks for this comment. We cited a new study regarding the prevalence/frequency of IgAN in this world [CJASN April 2017, 12 (4) 677-686].

2. P9 line 13 controlled trial

Sorry about this typo.

3. P10 eGFR>60 ml/min/1.732m2; should be eGFR<60 ml/min/1.732m2

Sorry about this typo.

Reviewer #2:

I have no competing interests.

Interesting study with some limitations.

Thanks for this comment. 

Abstract: why authors in the conclusion spoke about "Aggressive immunosuppressants should not

be given for all crescent IgAN and should be individualized". This does not seem to have relation with reported results

That’s only based on indirect evidence. We revised that as “Clinicians should be more careful to care of patients with crescent IgAN. Thanks for this comment.

Abstract. CI should be added for all dependent variables

We added all 95% CI. Thanks for this comment. 

Methods: normality should be checked for

OK. 

Methods: levels of significance for values inserted in multvariate analysis should be added

A Cox proportional hazard regression (shown as HR, 95% confidence interval (CI)) was used to analyze the possible factors for patient survival (both the univariate and multivariate Cox models).

Methods/results: ratio of n of events/n of variavbles seems to low (see PMID: 8970487)

Thanks for this comment. For EPV (number of events per variable) values of 10 or greater, no major problems occurred (J Clin Epidemiol . 1996 Dec;49(12):1373-9). EPV is 12/8, which is less than 10. However, in this study, the mean duration of follow-up was 7.2 ± 3.1 years. As for study regarding patient mortality, the possible bias maybe inevitable. After meticulous statistical analysis (Cox regression model, univariate and multivariate), the hazard ratio is significant. The result is consistent to previous studies. Therefore, we believed that the result of this study is true. Thanks for this comment.

---

## [Decision Letter · Decision Letter 1]

6 Jul 2020

PONE-D-20-15033R1

Crescents formations are independently associated with higher mortality in biopsy-confirmed immunoglobulin A nephropathy

PLOS ONE

Dear Dr. Tsai,

Thank you for submitting your manuscript to PLOS ONE. After careful consideration, we feel that it has merit but does not fully meet PLOS ONE’s publication criteria as it currently stands. Therefore, we invite you to submit a revised version of the manuscript that addresses the points raised during the review process.

Please revise Kidney biopsies according to the Oxford classification and try to address all other issues raised by Referee 1.

We look forward to receiving your revised manuscript.

Kind regards,

Fabio Sallustio

Academic Editor

PLOS ONE

Reviewers' comments:

Reviewer's Responses to Questions

2. Is the manuscript technically sound, and do the data support the conclusions?

Reviewer #1: Partly

3. Has the statistical analysis been performed appropriately and rigorously? 

Reviewer #1: N/A

4. Have the authors made all data underlying the findings in their manuscript fully available?

Reviewer #1: No

5. Is the manuscript presented in an intelligible fashion and written in standard English?

Reviewer #1: No

6. Review Comments to the Author

Reviewer #1: This referee invite the authors to revise the manuscript according to the suggestions listed in the first

comments, otherwise the paper will be rejected by this referee.

Kidney biopsies must be revised according to the Oxford classification described by Haas et al in JASN 28:691-701, 2017.

7. PLOS authors have the option to publish the peer review history of their article (what does this mean?). If published, this will include your full peer review and any attached files.

Reviewer #1: No

---

## [Author Response · Author response to Decision Letter 1]

7 Jul 2020

We reviewed our data regarding renal biopsy, and we re-scored it by Oxford classification with MEST-C score. We replaced Haas classification with Oxford classification in table 1. We also revised some parts in text. Thanks for your suggestions. 

 Total (n=388) Non Crescents(n=343) Crescents (n=45) p value

Oxford classification (n, %) 

M0 247 223 (90.3%) 24 (53.3%) 0.078

M1 138 118 (34.4%) 20 (44.4%) 0.087

M2 3 2 (0.6%) 1 (2.2%) 0.227

E0 227 296 (86.3%) 31 (68.9%) 0.04

E1 62 48 (14.0%) 14 (31.1%) 0.002

S0 341 322 (93.9%) 19 (45.2%) <0.001

S1 47 21 (6.1%) 26 (57.8%) <0.001

T0 332 298 (86.9%) 34 (75.6%) 0.073

T1 38 29 (8.5%) 9 (20%) 0.028

T2 18 16 (4.7%) 2 (4.4%) 0.788

C0 388 343 (100%) 0 (0%) <0.001

C1 35 0 35 (77.8%) <0.001

C2 10 0 10 (22.2%) <0.001

---

## [Decision Letter · Decision Letter 2]

21 Jul 2020

Crescents formations are independently associated with higher mortality in biopsy-confirmed immunoglobulin A nephropathy

PONE-D-20-15033R2

Dear Dr. Tsai,

We’re pleased to inform you that your manuscript has been judged scientifically suitable for publication and will be formally accepted for publication once it meets all outstanding technical requirements.

Kind regards,

Fabio Sallustio

Academic Editor

PLOS ONE

---

## [Editor Report · Acceptance letter]

23 Jul 2020

PONE-D-20-15033R2 

Crescents formations are independently associated with higher mortality in biopsy-confirmed immunoglobulin A nephropathy 

Dear Dr. Tsai:

I'm pleased to inform you that your manuscript has been deemed suitable for publication in PLOS ONE. Congratulations! Your manuscript is now with our production department. 

Kind regards, 

on behalf of

Dr. Fabio Sallustio 

Academic Editor

PLOS ONE